☰ PLOS | ONE

# Myocardial global longitudinal strain: An early indicator of cardiac interstitial fibrosis modified by spironolactone, in a unique hypertensive rat model

Catherine J. Leader[1], Mohammed Moharram[1], Sean Coffey[1], Ivan A. Sammut[2], Gerard W. Wilkins[1], Robert J. Walker[1]*

1 Department of Medicine, University of Otago, Dunedin, New Zealand, 2 Department of Pharmacology, University of Otago, Dunedin, New Zealand

* rob.walker@otago.ac.nz

## Abstract

### Objectives

Is global longitudinal strain (GLS) a more accurate non-invasive measure of histological myocardial fibrosis than left ventricular ejection fraction (LVEF) in a hypertensive rodent model.

### Background

Hypertension results in left ventricular hypertrophy and cardiac dysfunction. Speckle-track-ing echocardiography has emerged as a robust technique to evaluate cardiac function in humans compared with standard echocardiography. However, its use in animal studies is less clearly defined.

### Methods

Cyp1a1Ren2 transgenic rats were randomly assigned to three groups; normotensive, untreated hypertensive or hypertensive with daily administration of spironolactone (human equivalent dose of 50 mg/day). Cardiac function and interstitial fibrosis development were monitored for three months.

### Results

The lower limit of normal LVEF was calculated to be 75%. After three months hypertensive animals (196±21 mmHg systolic blood pressure (SBP)) showed increased cardiac fibrosis (8.8±3.2% compared with 2.4±0.7% % in normals), reduced LVEF (from 81±2% to 67±7%) and impaired myocardial GLS (from -17±2% to -11±2) (all p<0.001). Myocardial GLS demonstrated a stronger correlation with cardiac interstitial fibrosis ($r^2$ = 0.58, p<0.0001) than LVEF ($r^2$ = 0.37, p<0.006). Spironolactone significantly blunted SBP elevation (184±15, p<0.01), slowed the progression of cardiac fibrosis (4.9±1.4%, p<0.001), reduced the

**Funding:** Funding was provided by the Department of Medicine University of Otago, New Zealand (CL), a Laurenson Award from the Otago Medical Research Foundation Otago, New Zealand (IS,RW), the Healthcare Otago Charitable Trust Otago, New Zealand (RW), the New Zealand Lottery Grants Board (IS, GW, RW) and the Maurice and Phyllis Paykel Trust New Zealand (CL, RW). The funders had no role in study design, data collection and analysis, decision to publish, or preparation of the manuscript.

**Competing interests:** The authors have declared that no competing interests exist.

decline in LVEF (72±4%, p<0.05) and the degree of impaired myocardial GLS (-13±1%, p<0.01) compared to hypertensive animals.

## Conclusions

This study has demonstrated that, myocardial GLS is a more accurate non-invasive measure of histological myocardial fibrosis compared to standard echocardiography, in an animal model of both treated and untreated hypertension. Spironolactone blunted the progression of cardiac fibrosis and deterioration of myocardial GLS.

## Introduction

Sustained hypertension frequently results in left ventricular hypertrophy (LVH). In its early stages, the hypertrophied ventricle is able to compensate for the increased after-load. This is associated with progressive remodelling of the myocardium consisting of myocyte hypertrophy, accumulation of fibroblasts and collagen formation. This eventually leads to a reduction in left ventricular (LV) compliance, diastolic dysfunction, and subsequent systolic dysfunction, resulting in left ventricular decompensation and finally, heart failure [1–4]. Traditionally, left ventricular ejection fraction (LVEF) has been utilised as the main prognostic indicator of cardiac dysfunction. However, it is becoming increasingly apparent that the prognosis of heart failure may not be easily assessed by the LVEF alone, particularly in those patients with preserved LVEF [5].

Left ventricular hypertrophy and cardiac dysfunction in humans is routinely examined by echocardiography and magnetic resonance imaging. Such non-invasive imaging techniques for assessing cardiac performance in animal models (especially in rodents) are more difficult, in part due to the exceptional spatial and temporal resolution required to image a small rapidly beating heart. Thus, although research studies involving rodents commonly make use of clinical echocardiography systems, they often rely on simple conventional measures derived from M-mode echocardiographic tracings, which suffer from a number of shortcomings [6–8]. This, in turn, has led to difficulties in the assessment of diastolic function and contractile function, limiting the ability of animal models to replicate diastolic dysfunction or heart failure with preserved ejection fraction. To overcome these shortcomings, two-dimensional speckle-tracking echocardiography and the use of myocardial deformation (strain) analysis is emerging as a more robust technique to evaluate cardiac function [9]. Speckle tracking based strain analysis quantifies myocardial deformation by tracking the ultrasonographic motion of speckles throughout the cardiac cycle. To date, despite recent advances enabling this application in small animal models, the adoption of strain imaging in rodent models has been limited with very few published studies correlating the ultrasonographic changes with actual changes in cardiac interstitial fibrosis.

Since hypertension is a common and increasingly significant cause of mortality, a number of transgenic hypertensive rat models have been developed [10,11]. One such model is the transgenic Cyp1a1Ren2 rat, in which hypertension can be reversibly induced by diet, without the need for surgical intervention [12]. In this transgenic rat, mouse Ren2 cDNA expression is under the control of an inducible cytochrome p450-1a1 promoter, integrated into the Y chromosome of Fischer 344 rats [12,13], and is therefore only active in males. Dietary administration of indole-3-carbinol (I3C) leads to activation of the promoter gene (Cyp1a1) resulting in increased expression of Ren2 [12,14]. I3C is a naturally occurring, nontoxic, xenobiotic found

in cruciferous vegetables (such as broccoli) that acts as a benign inducer with a short half-life. Activation of Ren2, primarily in the liver, upon induction of the Cyp1a1 promoter by I3C [15,16], leads to increased circulating renin levels, activation of the renin-angiotensin-aldosterone system and a consequent increase in blood pressure. Significantly, the extent of hypertension is also I3C dose dependent [13,17,18], allowing for tight titration of blood pressure.

The role of mineralocorticoid receptor antagonists (MRA), such as spironolactone, in modulating the actions of aldosterone has been investigated in a number of animal studies [19–24] as well as in clinical trials [25–28], and a number of reviews outline their beneficial effects [29–33]. Most animal studies investigating the actions of MRAs examined their impact on cardiac remodelling after cardiac injury [19–22]. Only a few studies have investigated the effects of MRA in hypertensive models [19,22,23,34,35]. In these studies, spironolactone was administered (often at supra-physiological doses) at the onset or prior to onset of hypertension, which does not reflect the clinical setting of hypertension-mediated injury. In addition, the observation period was relatively short. Spironolactone reduced the rate of development of cardiac fibrosis, which is thought to be both dependent and independent of the actions of angiotensin II [30,36–38]. So far, to our knowledge, there are no animal studies examining the effect of chronic spironolactone therapy, at relevant clinical doses, following the establishment of hypertension and cardiac hypertrophy.

Therefore, the aim of this study was to investigate whether spironolactone modifies both cardiac functional parameters and cardiac interstitial fibrosis over time, following the establishment of progressive hypertensive injury, more accurately representing the clinical setting. Cardiac functional status was assessed by standard serial echocardiography with the addition of two-dimensional speckle-tracking echocardiography to assess global longitudinal strain (GLS) and these changes were compared with the histological evidence of cardiac interstitial fibrosis.

## Methods

### Animals

The initial transgenic Cyp1a1Ren2 rat internal breeding stock was gifted by Professor J.J. Mullins (Centre for Cardiovascular Science, University of Edinburgh, UK). The transgenic Cyp1a1Ren2 rat colony was held at the University of Otago Animal Resource Unit and animals were housed under controlled conditions of temperature (~21˚C) and light (12-h light/dark cycle), with food (meat-free rat and mouse diet, irradiated, Specialty Feeds, Australia) and tap water provided *ad libitum*. All Cyp1a1-Ren2 rats used for experiments were obtained from internal breeding stock and housed in pairs or in groups of four per cage. All experiments were approved by the Animal Ethics Committee of the University of Otago (AEC 51/13), in accordance with the guidelines of the New Zealand Animal Welfare Act [18, 39].

**Chronic elevation of blood pressure.**   Eight-week-old male transgenic Cyp1a1Ren2 rats (n = 28) were maintained on either irradiated pelleted standard chow (Meat free rat and mouse diet, Specialty Feeds, Perth, Australia) or irradiated pelleted standard chow with addition of 0.167% w/w indole-3-carbinol (I3C) (#SF13-086, Specialty Feeds, Perth, Australia) to activate hypertension [18]. Hypertension was established over two weeks (until 10 weeks of age) [18]. Animals were then randomised to either a hypertensive group (H) or a hypertensive group with spironolactone (H+SP). At four weeks, a subset of animals from both group H (n = 4) and group H+SP (n = 4) were euthanised for histological examination. The remaining animals (hypertensive (H, n = 8), hypertensive plus spironolactone (H+SP, n = 4), together with the untreated normotensive group (N, n = 8)) were euthanised at the end of the 12 week experimental period. Systolic blood pressure (SBP), weight and echocardiography were

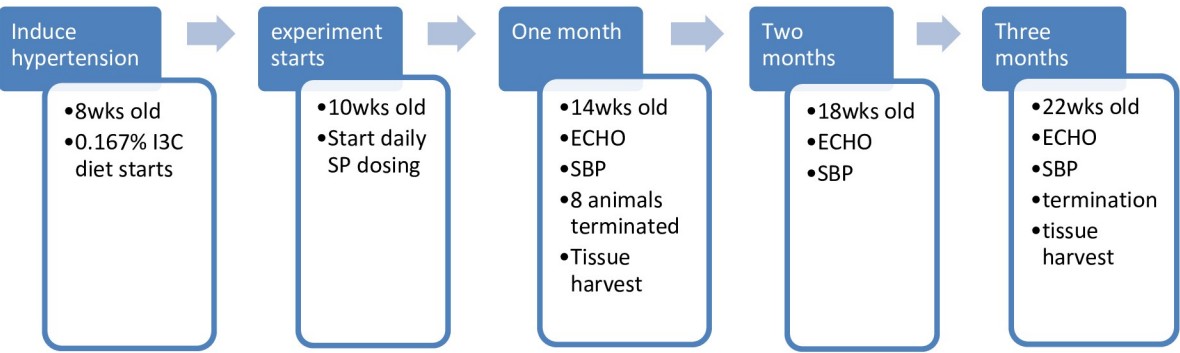

**Fig 1. Overview of the experimental design.** Animals were given an initial two weeks on the 0.167% w/w indole-3-carbinol (I3C) diet to establish hypertension before commencing daily spironolactone (SP) dosing. Every month, animals had systolic blood pressure (SBP) measured and an echocardiogram (Echo) performed. After one month, a sub set of animals (n = 8) were terminated, while the remaining animals were terminated following three months. Normotensive animals were only terminated after three months.

recorded regularly throughout the 12 week period on these animals ([Fig 1]). All animals were terminated by halothane overdose followed by cardiac puncture.

**Spironolactone dosing.** Dosing for these animals was adjusted using the Food and Drug Administration's 2005 allometric scaling calculations as described by Reagan-Shaw et al [40]. A human equivalent dose of 50mg per day of spironolactone (Sigma-Aldrich, Missouri, USA) was used, which equated to a dose of 4.41mg/kg/day for a rat. To enhance acceptance, spironolactone was mixed into a caramel syrup (Quaterpast, Shott Beverages Ltd., Auckland, New Zealand).

**Systolic blood pressure.** All rats were gentled by daily handling and weighed before commencing the experimental protocols, and weekly thereafter. Systolic blood pressure (SBP) was measured every four weeks in habituated rats after light sedation with midazolam (1.5mg/kg, *I.P.*), using tail-cuff plethysmography (NIBP controller plus PowerLab 4SP, ADInstruments, Dunedin, New Zealand). Animals were given 30 minutes to acclimatise prior to the blood pressure recording procedure and a heat lamp was used to gently warm the tail prior to SBP readings [18]. Data was captured and analysed using Chart v.7 software (ADInstruments, Dunedin, New Zealand). A mean of ten clear recordings were taken from each rat on each occasion.

**Echocardiography.** Echocardiography was performed every four weeks. Rats were anesthetised (5% isoflurane in oxygen 1L/min), maintained on 2% isoflurane in 1L/min oxygen and placed supine on an electrical heating pad (to maintain body temperature). The animal's chest was shaved and transthoracic echocardiography was conducted using a 10MHz linear probe (GE ML6-15, GE Healthcare, Chicago, USA) connected to a commercially available echocardiography system (Vivid 9, GE Healthcare, Chicago, USA). Standard two-dimensional and M-mode long- and short-axis (at the mid-papillary level) images were acquired. At least three consecutive cardiac cycles were acquired and transferred for offline analysis using an image analysis package (2D CPA, TomTec Image-Arena, version 2.21; TomTec Imaging Systems, Unterschleissheim, Germany). Determination of LV ejection fraction (LVEF) and end systolic volume were performed using Simpson's method on TomTec Image-Arena. Additionally, LVEF was measured using standard techniques using both two-dimensional and M-mode images in the same animals (EchoPac software version.112.0.x, GE Healthcare, USA) for comparison.

**Speckle-tracking echocardiography.** Loops of long axis views were acquired for speckle-tracking analysis, each using a frame rate of 97 frames/s (>15 frames/cardiac cycle). Analysis was performed in a blinded fashion by two experienced operators (MM, SC), using the

speckle-tracking algorithm incorporated into the TomTec analysis package (TomTec Image-Arena, TomTec Imaging Systems, Unterschleissheim, Germany). Standard long axis view was used for the analysis of GLS of the left ventricle. The GLS was measured at end-systole and averaged over three cardiac cycles. The timings of end-systole and end-diastole were determined by M-Mode analysis of aortic and mitral valves. In the defined end-systolic frame the endocardial border of the left ventricle was traced manually starting at one end of the mitral annulus through the entire endocardium and ending where the aortic annulus joins the left ventricular myocardium in the long axis view. The analysis software generated a region of interest (ROI) including the entire myocardial thickness. The width of the ROI was manually adjusted as required to include the entire myocardium and exclude the pericardium. The tracking quality was then assessed to verify its adequacy across the cardiac cycle. The software calculates GLS using the entire myocardial line length according to the latest recommendations for assessing myocardial deformation [41], with layered strain allowing separate measurement of myocardial GLS (myo-GLS) and endocardial GLS (endo-GLS), referring to tracking of the mid-myocardial and endocardial layers, respectively. The inter-observer and intra-observer interclass variability correlation coefficients were 96% and 97% respectively.

**Quantitative microscopy (histology).** Hearts were removed, placed into Hartman's saline (with 15mmol.L$^{-1}$ potassium chloride) to arrest the heart in diastole, and fixed for four hours in 10% neutral buffered formalin (NBF). The heart was transversely cut into 3mm sections, measuring from the apex using a heart matrix, before further fixation in 10% NBF overnight at room temperature. Sections were then dehydrated by passage through alcohols and embedded in paraffin wax. Cardiac sections were cut at 5μm and stained with picrosirius red with light green counter stain.

Stained sections were viewed using a Zeiss Axioplan Microscope (Zeiss, Oberkochen, Germany), and images of representative regions (0.34mm$^2$) were recorded using a Nikon microscope camera (DS-Ri2, Nikon, Tokyo, Japan). Cardiac interstitial fibrosis was quantitatively assessed using Sirius red, capturing a minimum of 10 non-overlapping, evenly distributed, myocardial sections (x50 magnification), containing no vessels, from the transverse section taken 6mm from the apex from each animal. The extent of fibrotic tissue was quantified by applying a trained pixel classifier (NIS Elements Basic Research Imaging software, Version 5.11 (64bit edition), Nikon, Tokyo, Japan) to each section (as a percentage of the total image) and further averaged for individual animals and then each group.

Use of the section 6mm from the apex was justified by calculating the estimated total volumetric fibrosis of the heart (up to 9mm from the apex) from a subset of animals and comparing that to the fibrosis percentage obtained from the 6mm section alone (see S1 Fig, S1 Table).

## Statistics

Quantitative data are presented as mean ± standard deviation. Statistical comparisons were accomplished by unpaired Student's t-test or one-way analysis of variance (ANOVA) with Bonferroni post-hoc analysis and correlations were performed using Pearson's correlation (GraphPad Prism, GraphPad software, Inc. version 5.03). Results were considered to be statistically significant if P values were <0.05.

## Results

All rats showed a steady weight gain over the three month experimental period. Normotensive rats maintained a consistent heart rate and systolic blood pressure (SBP, 80–104 mmHg), left ventricular ejection fraction (LVEF, >78.5%) and GLS (endo-GLS, <-24.7%, myo-GLS, <-15.8%) over the study duration. (Table 1, Fig 2). Using a 95% confidence interval of the data

**Table 1. Physiological measurements.**

| | | N | H | H+SP |
|---|---|---|---|---|
| **SBP (mmHg)**[a] | 1mth | 94±7 [89–99] | 172±14 [163–182]*** | 181±23 [159–204]*** † |
| | 2mth | 94±8 [89–100] | 188±21 [174–203]*** | 185±16 [179–201]*** |
| | 3mth | 91±13 [82–101] | 196±21 [182–210]*** | 184±15 [170–198]*** †† |
| **LVEF (%)**[b] | 1mth | 84±4 [81–87] | 81±2 [80–83] | 83±2 [81–85] |
| | 2mth | 84±2 [83–86] | 73±4 [70–75]*** | 77±2 [74–79]*** †† |
| | 3mth | 80±3 [78–82] | 67±7 [68–72]*** | 72±4 [68–75]*** † |
| **End systolic volume (µl)** | 1mth | 50±18 [6–94] | 53±9 [46–61] | 45±7 [34–56] |
| | 2mth | 43±2 [41–46] | 77±8 [70–83]*** | 72±10 [56–87]** |
| | 3mth | 58±10 [49–68] | 86±11 [77–95]*** | 80±10 [64–96]* |
| **Endocardial GLS (%)**[c] | 1mth | -30±2 [-33 - -28] | -29±3 [-31 - -27] | -31±2 [-32 - -29] |
| | 2mth | -30±2 [-31 - -29] | -22±2 [-23 - -21]*** | -26±1 [-27 - -24]*** ††† |
| | 3mth | -27±3 [-29 - -24] | -19±3 [-21 - -17]*** | -21±2 [-23 - -19]*** † |
| **Myocardial GLS (%)**[c] | 1mth | -19±2 [-21 - -18] | -17±2 [-17.9 - -15]** | -19±2 [-20 - -17]†† |
| | 2mth | -19±2 [-20 - -17] | -14±2 [-15 - -13]*** | -15±1 [-16 - -14]*** |
| | 3mth | -17±2 [-19 - -15] | -11±2 [-12 - -9]*** | -13±1 [-14 - -12]*** †† |

Physiological measurements of normotensive (N), hypertensive (H) and hypertensive rats dosed daily with spironolactone (H+SP) following one, two and three months (n = 4–8 per group). Values are shown as mean ± standard deviation, with 95% confidence intervals shown in brackets.

[a] Systolic blood pressure (SBP) was measured via tail cuff.

[b] Left ventricular ejection fraction (LVEF) and end systolic volume were calculated from echocardiograms.

[c] Endocardial and myocardial global longitudinal strain (GLS) was determined from speckle tracking.

* indicates significantly different from N. * $p < 0.05$,

** $p < 0.01$,

*** $p < 0.001$

† indicates significance between H and H+SP. † $p < 0.05$,

†† $p < 0.01$,

††† $p < 0.001$

obtained from the normotensive animals, a reference range was established for LVEF (>75%), cardiac fibrosis (<3%) and myo-GLS (<-15%). As there was no differences in the physiological data in normotensive animals at each time point, normotensive hearts at three months were used as the histological reference range.

Hypertensive rats showed a significant and rapid increase in SBP in the two weeks prior to the experimental starting point (from 93±10 mmHg to 160±17, p<0.001), as previously reported [18]. Systolic blood pressure continued to rise, subsequently reaching 172±14mmHg after one month and 196±21mmHg after three months. Associated with this progressive hypertension, LVEF gradually declined at each measured time point, along with a significant increase in end systolic volume and significant deterioration of GLS over the three month period (Table 1, Fig 2).

Hypertensive animals treated with spironolactone demonstrated a similar rise in SBP after one month compared to the untreated hypertensive group (181±23 vs 172±14mmHg, p<0.05). However, the subsequent rise in SBP seen in the hypertensive group was blunted over three months by spironolactone (196±21mmHg vs 184±15mmHg, respectively, p<0.01) (Table 1). This was associated with a significantly improved ejection fraction (Fig 2) and GLS after three months compared to the hypertensive group, although LVEF and GLS were still significantly reduced compared to the normotensive group (Table 1).

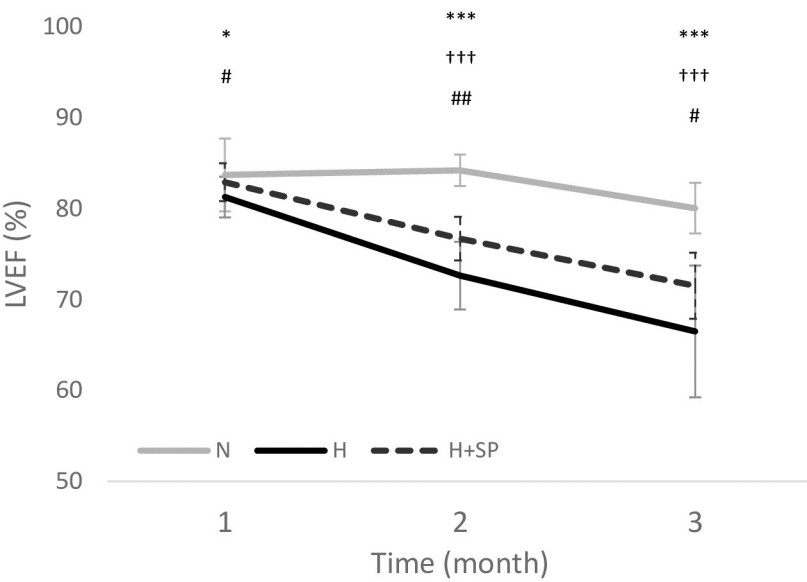

**Fig 2. Left ventricular ejection fraction after one, two and three months.** Left ventricular ejection fraction (LVEF) of normotensive (N, grey, n = 8), hypertensive (H, black, n = 8) and hypertensive animals dosed daily with spironolactone (H+SP, dashed line, n = 4) after one, two and three months. Values are shown as mean ± 95% confidence intervals. Significance between N and H is indicated by *, p<0.05 *, p<0.001 *** Significance between N and H+SP is indicated by †, p<0.001 ††† Significance between H and H+SP is indicated by #, p< 0.05 #, p<0.01 ##.

## Myocardial fibrosis

Fibrosis was significantly increased in hypertensive animals at both one month (4.4±2%) and three months (8.8±3.2%), when compared to normotensive animals (2.4±0.7%, p<0.001) (Fig 3). After one month, spironolactone treatment produced a significant blunting in fibrotic deposition when compared to untreated hypertensive group (3±1.2% vs 4.4±2%, p<0.01). After three months of spironolactone treatment, the progression of fibrosis remained significantly blunted compared to the untreated hypertensive group (4.9±1.4% vs 8.8±3.2%, p<0.001) (Fig 3).

## Global longitudinal strain

The normotensive group displayed relatively consistent endo-GLS and myo-GLS throughout the three month period. Endo-GLS was unchanged in the hypertensive group following one month of elevated SBP, but deteriorated by two months, leading to significantly reduced endo-GLS after three months when compared to the normotensive group (p<0.001) (Table 1). Significant impairment of myo-GLS was evident in the hypertensive animals after one month (p<0.01), and became more marked over the next two months (-11±2% vs -17±2% compared with the normotensive group, p<0.001). In contrast, the hypertensive group treated with spironolactone demonstrated no significant difference in myo-GLS from the normotensive group at one month, and although myo-GLS declined over the following two months, myo-GLS remained significantly better than in the untreated hypertensive animals (p<0.001) (Table 1, Fig 4).

## Correlation of ejection fraction and global longitudinal strain with myocardial fibrosis

At one month, hypertensive animals maintained the same LVEF and endo-GLS as normotensive animals, despite a significant increase in myocardial fibrosis. By three months, despite a

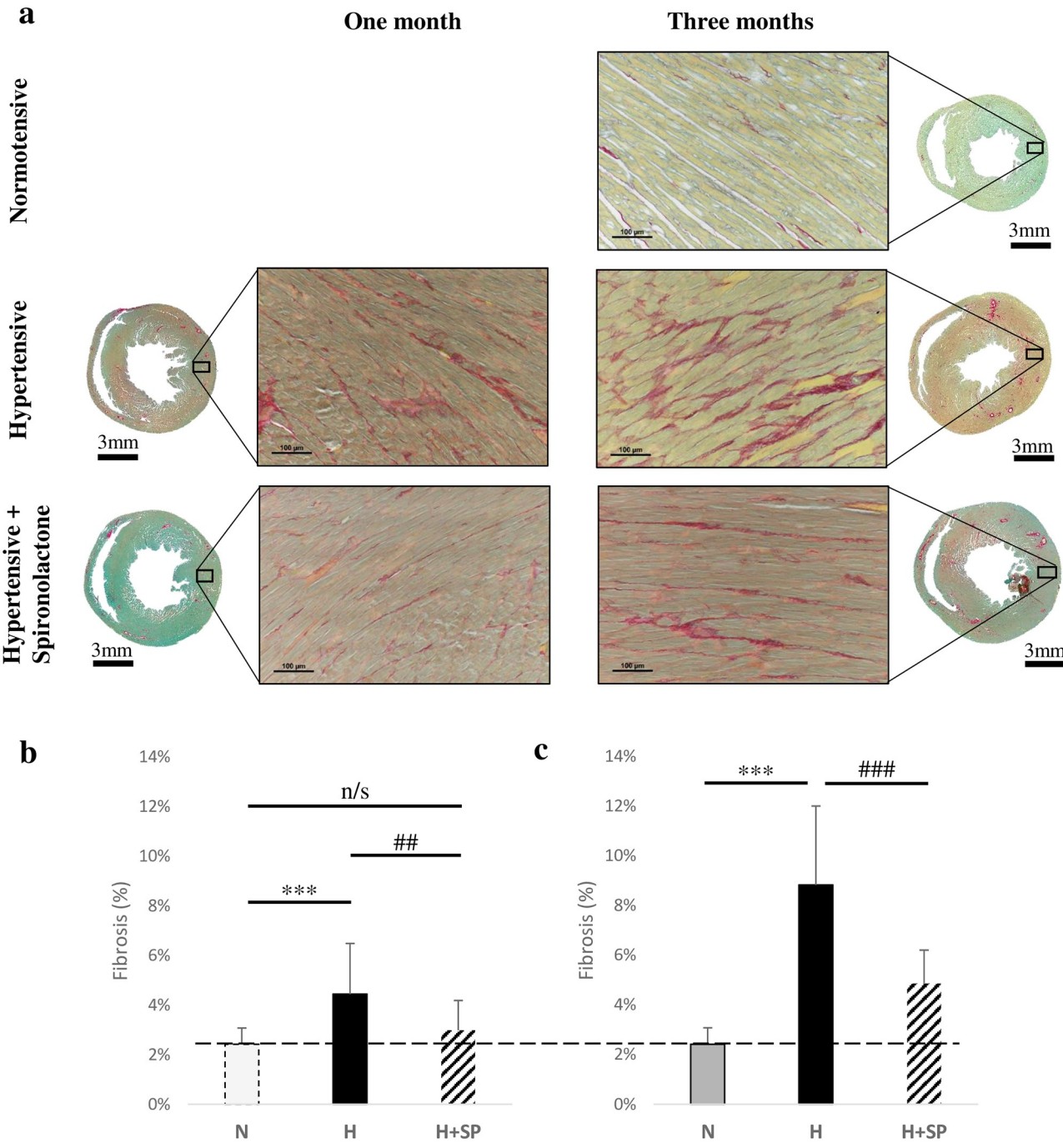

**Fig 3. Hypertension and cardiac fibrosis. 3(a)** Cardiac transverse sections, taken 6mm from the apex (scale bar is 3mm) from normotensive animals (N) after three months, hypertensive animals (H) and hypertensive animals with spironolactone (H+SP) after both one and three months. Sections were stained with picrosirius red and light green counter stain. Inlay sections (50x magnification, scale bar 100μm) are taken from the lateral wall of the left ventricle. **(3b, 3c)** Percentage of fibrosis of the left ventricle (transverse sections taken 6mm from the apex) from normotensive (N, grey, n = 8), hypertensive (H, black, n = 8) and hypertensive animals dosed daily with spironolactone (H+SP, striped, n = 4) following one month **(b)** and three months **(c)**. Values for normotensive animals were taken at three months only (dashed line), and were compared with hypertensive and spironolactone treated animals at both one month and at three months. Values are shown as mean ± standard deviation.

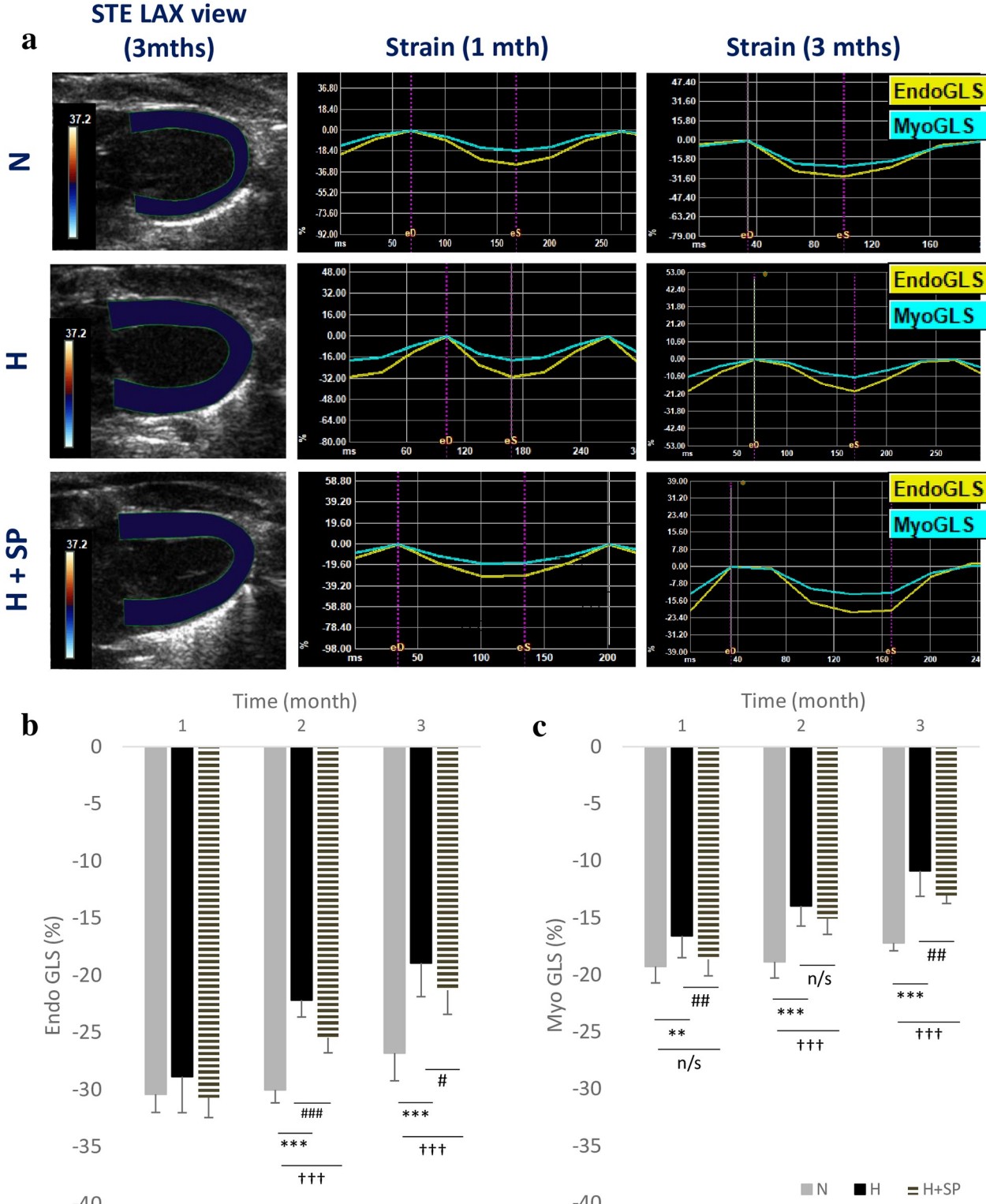

**Fig 4. Global longitudinal strain analysis.** Global longitudinal strain (GLS) tracking analysis **(a)**, showing the region of interest and the strain assessment at both one month and three months in normotensive (N), hypertensive (H) and hypertensive with spironolactone (H+SP) groups. **(b)** Analysis of the endocardium layer and **(c)** myocardium layer from normotensive (N), hypertensive (H) and hypertensive animals with spironolactone (H+SP) following one, two and three months. Values are shown as mean ± standard deviation. Significant differences between N and H is indicated by *, p<0.05 *, p<0.01 **, p<0.001 *** Significant differences between N and H+SP is indicated by †, p<0.05 †, p<0.01 †† Significant differences between H and H+SP is indicated by #, p< 0.05 #, p<0.01 ##.

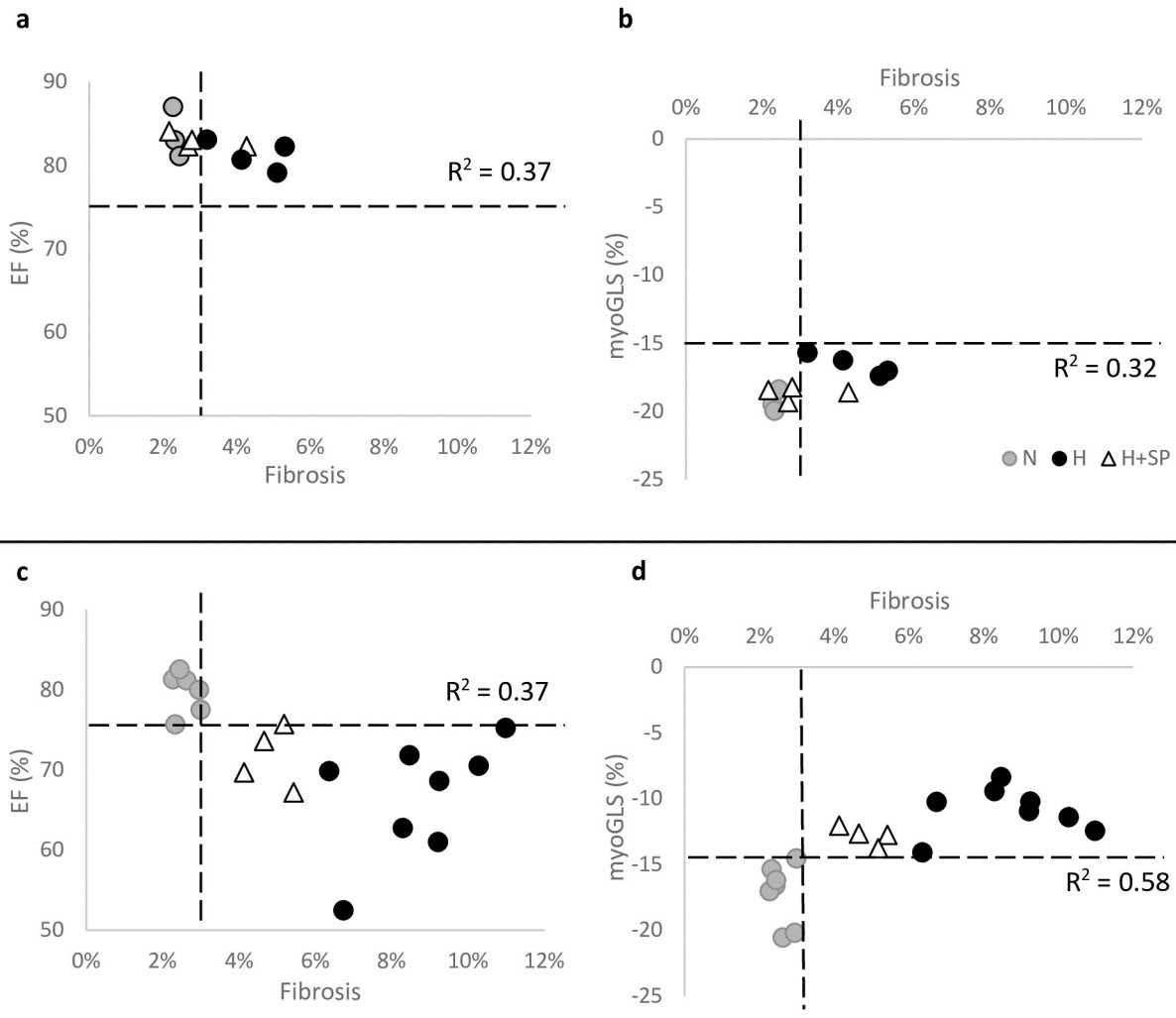

**Fig 5. Comparison of left ventricular fibrosis and myocardial global longitudinal strain.** Data from individual animals comparing left ventricular cardiac fibrosis (%) against EF% (**a, c**) and myo-GLS (**b, d**) in normotensive animals (N, grey circles, data only available following three months), hypertensive animals (H, black circles) and hypertensive animals dosed daily with spironolactone (H+SP, triangles) after one month (**a, b**) and three months (**c, d**). The black dotted lines represent the 95% CI normal reference range of the normotensive animals. $r^2$ correlation values are shown on each graph for all data.

further substantial increase in myocardial fibrosis, the measured ejection fraction was quite variable, but the myo-GLS was significantly impaired in the hypertensive animals (Table 1, Fig 5). Treatment with spironolactone significantly reduced the extent of fibrosis at one month, with preserved ejection fraction and preserved endo- and myo-GLS compared to normotensive animals. By three months, spironolactone continued to limit fibrosis development which was associated with a significant reduction in impairment of myo-GLS, and to a lesser extent LVEF, when compared to hypertensive animals at three months. (Table 1, Fig 5).

Linear correlations with myocardial fibrosis measurements at one month revealed similar correlations of LVEF and myo-GLS ($r^2 = 0.37$, $p<0.047$, and $r^2 = 0.32$, $p = 0.06$, respectively). However, at three months myo-GLS revealed a much stronger correlation with myocardial fibrosis ($r^2 = 0.58$, $p<0.0001$) compared to LVEF ($r^2 = 0.37$, $p<0.01$) (Fig 5). Additionally, linear correlations of myocardial fibrosis and endo-GLS were not significant at one month ($r^2 = 0.07$, $p = 0.43$) but were significant at three months ($r^2 = 0.6$, $p<0.001$).

Standard assessments of LVEF compared with histological evidence of myocardial fibrosis, at one month M-mode or Simpson's rule did not show any significant correlation. After three months however, the difference in LVEF calculated by Simpson's rule (p = 0.006) was significant, whilst M-mode showed no significant correlation (p = 0.64) (S2 Table).

Raw data is attached as S1 Dataset.

## Discussion

In this animal model of inducible hypertension we have demonstrated that sustained severe hypertension resulted in significant progressive cardiac interstitial fibrosis. Spironolactone blunted the progression of cardiac fibrosis and deterioration of myocardial GLS after the establishment of hypertension. We have demonstrated that speckle tracking derived myocardial global longitudinal strain correlates more closely with changes in myocardial interstitial fibrosis compared to left ventricular cardiac ejection fraction derived from 2D echocardiography.

Echocardiography has remained the principal non-invasive imaging modality to provide reliable assessment of cardiac function. While left ventricular systolic function measured by M-mode, fractional shortening or 2D ejection fraction, has been shown to be a useful volumetric-based index, these measurements are limited by inherent variability, influenced by the quality of the image, off-axis imaging, measurement errors or by geometric confounders. Speckle tracking derived strain is still reliant on good quality, on-axis images, but is a direct measure of myocardial function rather than calculation of cardiac volumes. In recent clinical studies, the use of speckle tracking has shown GLS as a diagnostic tool to be superior to left ventricular ejection fraction (LVEF) in a wide range of cardiac conditions [42–45], especially in detecting subtle impairment in left ventricular function [44]. Additionally, these studies also report a significant reduction in GLS without a corresponding reduction in LVEF [42–45]. Strain analysis, particularly GLS, has also been reported clinically as a more sensitive predictor of overall CV mortality compared to LVEF [44, 46], and defined progressive changes in strain predicted mortality while changes in EF did not, until cardiac contractility was severely impaired. Furthermore these changes in GLS are similar in rats and humans [47]. In this study, we have shown that, impairment in myo-GLS is apparent even without detectable changes in LVEF in early stage disease, and correlates better with histological assessment of myocardial fibrosis in more advanced disease.

A separate issue is the use of human reference ranges for cardiac function in animals. Whilst the LVEF in the hypertensive group was reduced compared to normotensive animals, the LVEF values obtained would be considered as relatively normal (or preserved) based on the standard clinical definition of preserved LVEF being greater than 50% [5]. Our method of construction of a reference range based on control animals identified all diseased animals (with or without treatment with spironolactone) as having reduced LVEF by three months. As such, after three months of sustained hypertension, these animals would not be suitable as a model of heart failure with preserved ejection fraction (HFpEF) despite having LVEF > 50%. In contrast to other recommendations (2), we recommend construction of study-specific normal reference ranges for LVEF when performing preclinical research in HFpEF, as these are likely to significantly alter the cut-off for preserved LVEF. Notably, at 1 month, hypertensive animals in our study did have a preserved LVEF, but an impaired myo-GLS.

Recent studies performed with Dahl salt-sensitive rats [1,9] showed a similar gradual reduction in LVEF with a matched steep decline in GLS to that seen in our study. However, in that study, the animals had profoundly more severe SBP recorded (>220mmHg following 16 weeks of age [1]). This model provides histological support for the clinical survival

observations [46] where small changes in GLS present in patients with EFs still in the normal range are likely to have significant pathological changes present.

A number of studies have demonstrated a central role of aldosterone in promoting cardiac fibrosis [48–53], and have further established that this can be independent of blood pressure [52,54,55]. Early work by Brilla and colleagues [54], showed that aldosterone stimulated collagen synthesis through mineralocorticoid receptors in isolated cardiac fibroblasts. This work led to multiple animal studies showing that mineralocorticoid receptor antagonist (MRA) can prevent or delay the development of ventricular remodelling and cardiac interstitial fibrosis particularly following cardiac injury [33,36,51,56,57]. Clinical studies have also identified the potential beneficial roles of mineralocorticoid receptor antagonism in cardio-protection [25,26,29,57–59]. Unlike previously reported animal studies utilising MRAs, in this study we examined the effects of mineralocorticoid receptor antagonism on cardiac injury/fibrosis after establishment of severe hypertension. Following one month of spironolactone therapy, LVEF and myocardial GLS were maintained despite the lack of an apparent reduction in SBP. This was associated with a reduction in cardiac interstitial fibrosis compared to hypertensive animals. At three months of spironolactone therapy, although there was some progression in the extent of cardiac interstitial fibrosis along with a reduction in myocardial GLS and LVEF, this was not as marked as that seen in the untreated hypertensive animals (Fig 5).

There are a number of limitations to this study. While spironolactone is known to influence a number of pathophysiological cardiac effects [56], it is difficult to tease out from our data, whether the apparent reduction in the progression of cardiac fibrosis is due to direct actions of the MRA, the stabilisation in the rise of SBP, or both. Additionally, this study used the underlying assumption that the activation of the mineralocorticoid receptor, via increased RAAS activation, directly effects cardiac fibrosis.

Due to a lack of difference in the physiological data between one month and three months, only tissue from normal hearts at three months was used as the reference range. It is possible that a small degree of myocardial fibrosis related to age alone was missed, but given the significant differences between hypertensive and normotensive animals obtained, this would have little or no impact on the results presented. Due to the small heart size and rapid heart rate, it was not possible to assess diastolic function. Likewise, due to the small size of the hearts, the boundaries of the cardiac layers were at times difficult to distinguish, preventing an accurate assessment of the epicardial or endocardial layer. Further work with this rat model utilising an echocardiographic probe with the ability to capture at a higher frame rate would help establish both the systolic and diastolic cardiac dysfunction more accurately.

Spironolactone, given to rats following established severe hypertension, reduced the extent of cardiac interstitital fibrosis. Of note, by using both a human equivalent dose of spironolactone and oral dosing, as well as ensuring hypertension was established prior to therapeutic intervention; this more closely mimics the clinical setting, allowing for more accurate translation to clinical outcomes.

In summary, spironolactone blunted the progression of cardiac fibrosis and deterioration of myocardial GLS after the establishment of hypertension. Myocardial GLS (as opposed to LVEF or endocardial GLS) was more sensitive in detecting the early stages of hypertension-mediated cardiac injury, where ejection fraction is preserved, and had a closer correlation with histological myocardial fibrosis in late stage disease. These findings would suggest that measurement of myocardial GLS should be utilised in preference to M-mode and estimates of ejection fraction, providing increased statistical power and the ability to non-invasively assess myocardial fibrosis. We suggest that strain analysis should be more commonly used, in preclinical studies to allow better correlation with clinical analyses. Additionally, we recommend construction of study-specific normal reference ranges for LVEF when performing preclinical

research in HFpEF as these are likely to significantly vary significantly from the clinical definition for preserved LVEF, and hence the translational ability of the experimental findings. Further work is planned to categorise the cardiac dysfunction in this animal model and the potential mechanisms that may mediate the actions of spironolactone.

## Supporting information

**S1 Dataset. Raw data.**
(XLSX)

**S1 Fig. Calculation of total cardiac area.**
(DOCX)

**S1 Table. Total cardiac area and fibrosis compared to single mid-section (6mm from apex).**
(DOCX)

**S2 Table. Left ventricular ejection fraction obtained by different standard measurement techniques and correlation with interstitial fibrosis.**
(DOCX)

## Author Contributions

**Conceptualization:** Ivan A. Sammut, Gerard W. Wilkins, Robert J. Walker.

**Formal analysis:** Catherine J. Leader, Mohammed Moharram, Sean Coffey, Gerard W. Wilkins.

**Funding acquisition:** Ivan A. Sammut, Gerard W. Wilkins, Robert J. Walker.

**Investigation:** Catherine J. Leader, Robert J. Walker.

**Methodology:** Catherine J. Leader, Mohammed Moharram, Sean Coffey, Gerard W. Wilkins.

**Project administration:** Catherine J. Leader, Robert J. Walker.

**Resources:** Catherine J. Leader, Ivan A. Sammut, Robert J. Walker.

**Software:** Mohammed Moharram, Sean Coffey.

**Supervision:** Sean Coffey, Ivan A. Sammut, Gerard W. Wilkins, Robert J. Walker.

**Writing – original draft:** Catherine J. Leader.

**Writing – review & editing:** Catherine J. Leader, Mohammed Moharram, Sean Coffey, Ivan A. Sammut, Gerard W. Wilkins, Robert J. Walker.

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
