## [Decision Letter · Decision Letter 0]

5 Jul 2019

PONE-D-19-14918

Myocardial global longitudinal strain: an early indicator of cardiac interstitial fibrosis in a unique hypertensive rat model.

PLOS ONE

Dear Professor Walker,

Thank you for submitting your manuscript to PLOS ONE. After careful consideration, we feel that it has merit but does not fully meet PLOS ONE’s publication criteria as it currently stands. Therefore, we invite you to submit a revised version of the manuscript that addresses the points raised during the review process.

ACADEMIC EDITOR: All issues raised by expert reviewers are required. The authors should limit speculations and highlight limitations of the study.

We would appreciate receiving your revised manuscript by Aug 19 2019 11:59PM. To enhance the reproducibility of your results, we recommend that if applicable you deposit your laboratory protocols in protocols.io, where a protocol can be assigned its own identifier (DOI) such that it can be cited independently in the future. For instructions see: http://journals.plos.org/plosone/s/submission-guidelines#loc-laboratory-protocols

We look forward to receiving your revised manuscript.

Kind regards,

Vincenzo Lionetti, M.D., PhD

Academic Editor

PLOS ONE

Journal Requirements:

https://www.ingentaconnect.com/content/aalas/cm/2018/00000068/00000005/art00005

In your revision ensure you cite all your sources (including your own works), and quote or rephrase any duplicated text outside the methods section. Further consideration is dependent on these concerns being addressed.

4. To comply with PLOS ONE submissions requirements, please provide methods of sacrifice in the Methods section of your manuscript.

5. Data availability issue. In your statement you say "All relevant data are within the paper and its Supporting Information files", but as we explain in http://journals.plos.org/plosone/s/data-availability#loc-faqs-for-data-policy you should provide the individual data points behind means, medians and variance measures presented in the results, tables and figures, and not just those summary statistics. Please provide these underlying participant-level data in a supporting information file or public repository, taking care not to include identifying information (see http://www.bmj.com/content/340/bmj.c181.long); if these data cannot be publicly deposited or included in the supporting information, e.g. due to patient privacy, legal reasons, or being provided by a third party, please explain why and explain how researchers may access them. Note that authors should not be the sole named individuals responsible for ensuring data access.

Reviewers' comments:

Reviewer's Responses to Questions

**Comments to the Author**

1. Is the manuscript technically sound, and do the data support the conclusions?

Reviewer #1: Yes

Reviewer #2: No

2. Has the statistical analysis been performed appropriately and rigorously? 

Reviewer #1: Yes

Reviewer #2: Yes

3. Have the authors made all data underlying the findings in their manuscript fully available?

Reviewer #1: Yes

Reviewer #2: Yes

4. Is the manuscript presented in an intelligible fashion and written in standard English?

Reviewer #1: Yes

Reviewer #2: Yes

5. Review Comments to the Author

Reviewer #1: This article investigates the effect of spironolactone on cardiac function and fibrosis after induction of hypertension in a transgenic hypertensive rat model. The authors perfomes an advanced echocardiograpic evaluation assessing global longitudinal strain (myocardial and endocardial) that usually is not evaluated in the animal model.

As the authors report GLS can give more information in particular in HFpEF where ejection fraction is preserved.

The article is well written and the topic is relevant in clinical setting.

There are some suggestions:

- the authors should consider to add echocardiographic data of mass in Table 1:

- the discussion section could be a little bit shortened

- the are two figure 1 and no figure 2

- there are two figure 5 and no figure 4.

Reviewer #2: This experimental study using a hypertensive rat model showed the reduction of myocardial fibrosis with spironolactone, showing the changes in left ventricular ejection fraction and longitudinal strain from speckle tracking echo.

However, the findings do not provide novel insights as compared to previous studies referred.

Comments

1. The title does not imply the aim and results exactly. The effect of spironolactone should be reflected in the title.

2. Myo-GLS is more sensitive to the myocardial fibrosis, as shown in Fig 5. However, in the mid portion of the myocardium, cardiac fiber may be located in the circumferential positions rather than longitudinal positions. Please explain the discrepancy between functional and pathological behaviors.

3. In addition, the transmural distribution of fibrosis is needed to conclude your findings.

4. This technique may have limitations in reproducibility of strain measurements. The intra- and inter-observers’ variabilities should be clarified.

6. PLOS authors have the option to publish the peer review history of their article (what does this mean?). If published, this will include your full peer review and any attached files.

Reviewer #1: Yes: Flora Pirozzi

Reviewer #2: No

---

## [Author Response · Author response to Decision Letter 0]

12 Jul 2019

ACADEMIC EDITOR: All issues raised by expert reviewers are required. The authors should limit speculations and highlight limitations of the study.

Response: Comments to the reviewers comments are detailed below. Changes made to manuscript both in PLOS ONE style as well as where requested.

Reviewer #1: This article investigates the effect of spironolactone on cardiac function and fibrosis after induction of hypertension in a transgenic hypertensive rat model. The authors perfomes an advanced echocardiograpic evaluation assessing global longitudinal strain (myocardial and endocardial) that usually is not evaluated in the animal model.

As the authors report GLS can give more information in particular in HFpEF where ejection fraction is preserved.

The article is well written and the topic is relevant in clinical setting.

There are some suggestions:

- the authors should consider to add echocardiographic data of mass in Table 1:

- the discussion section could be a little bit shortened

- the are two figure 1 and no figure 2

- there are two figure 5 and no figure 4.

Response: We thank the reviewer for her helpful comments. LV mass was considered but as this is also a derived calculation, it did not add any additional information, to what was already observed. So it has not been added. The discussion has been shortened where possible (see track change version). The incorrect labelling of the figures has been changed.

Reviewer #2: This experimental study using a hypertensive rat model showed the reduction of myocardial fibrosis with spironolactone, showing the changes in left ventricular ejection fraction and longitudinal strain from speckle tracking echo.

However, the findings do not provide novel insights as compared to previous studies referred.

We would like to disagree with the reviewer. There are several important novel insights. Firstly we have reported the strong correlation between myo-GLS and the degree of cardiac interstitial fibrosis as seen histologically, confirming it as an important non-invasive means to assess cardiac fibrosis. This has not previously been reported. 

It is important to clearly define a normal range for ejection fraction in an animal model and not rely on a human range. We believe this is the first time it has been defined for an animal model. This also highlights the potential for this animal model as a model for HFpEF as identified by reviewer 1.

 Previous intervention studies have administered spironolactone (or other drugs) at the onset of hypertension, which does not reflect the clinical scenario where patients present with established hypertension and left ventricular hypertrophy. Our model more accurately reflects this. We used a dose for the rats equivalent to the standard human dose. Previous animal studies with spironolactone use supra-physiological doses (often 10 to 100 fold higher than what would be an equivalent human dose) and these are administered either subcutaneously (osmotic pumps) or intra-peritoneally, when spironolactone has no evidence for efficacy other than by oral administration as was used in this study. 

Comments

1. The title does not imply the aim and results exactly. The effect of spironolactone should be reflected in the title.

Thank you for this. The title has been changed appropriately.

2. Myo-GLS is more sensitive to the myocardial fibrosis, as shown in Fig 5. However, in the mid portion of the myocardium, cardiac fiber may be located in the circumferential positions rather than longitudinal positions. Please explain the discrepancy between functional and pathological behaviors. 

Myocardial global longitudinal strain measures contractile activity in real-time, and cardiac fibers are not static, rather they contract in a 3 dimensional response. This is picked up and allowed for in the global longitudinal strain analysis. Therefore any alterations in contractility related to fibrosis would still be observed. 

3. In addition, the transmural distribution of fibrosis is needed to conclude your findings.

Serial sections were analysed for the extent of fibrosis and we focused on the myocardial component rather than including epi and endocardial regions as these were too small to be of significance. Figure 3 demonstrates the regions of interest.

4. This technique may have limitations in reproducibility of strain measurements. The intra- and inter-observers’ variabilities should be clarified.

The echo analyses were undertaken blinded by two of the investigators (MM, SC). The inter-observer and intra-observer interclass variability correlation coefficients were 96% and 97% respectively.

---

## [Decision Letter · Decision Letter 1]

25 Jul 2019

Myocardial global longitudinal strain: an early indicator of cardiac interstitial fibrosis modified by spironolactone, in a unique hypertensive rat model.

PONE-D-19-14918R1

Dear Dr. Walker,

We are pleased to inform you that your manuscript has been judged scientifically suitable for publication and will be formally accepted for publication once it complies with all outstanding technical requirements.

With kind regards,

Vincenzo Lionetti, M.D., PhD

Academic Editor

PLOS ONE

Additional Editor Comments (optional):

Reviewers' comments:

Reviewer's Responses to Questions

**Comments to the Author**

1. If the authors have adequately addressed your comments raised in a previous round of review and you feel that this manuscript is now acceptable for publication, you may indicate that here to bypass the “Comments to the Author” section, enter your conflict of interest statement in the “Confidential to Editor” section, and submit your "Accept" recommendation.

Reviewer #1: All comments have been addressed

Reviewer #2: All comments have been addressed

2. Is the manuscript technically sound, and do the data support the conclusions?

Reviewer #1: Yes

Reviewer #2: Yes

3. Has the statistical analysis been performed appropriately and rigorously? 

Reviewer #1: Yes

Reviewer #2: Yes

4. Have the authors made all data underlying the findings in their manuscript fully available?

Reviewer #1: Yes

Reviewer #2: Yes

5. Is the manuscript presented in an intelligible fashion and written in standard English?

Reviewer #1: Yes

Reviewer #2: Yes

6. Review Comments to the Author

Reviewer #1: (No Response)

Reviewer #2: Authors politely addressed my questions and comments asked. The manuscript has been improved, then,I have no further comments.

7. PLOS authors have the option to publish the peer review history of their article (what does this mean?). If published, this will include your full peer review and any attached files.

Reviewer #1: Yes: Flora Pirozzi

Reviewer #2: No

---

## [Editor Report · Acceptance letter]

2 Aug 2019

PONE-D-19-14918R1 

Myocardial global longitudinal strain: an early indicator of cardiac interstitial fibrosis modified by spironolactone, in a unique hypertensive rat model. 

Dear Dr. Walker:

I am pleased to inform you that your manuscript has been deemed suitable for publication in PLOS ONE. Congratulations! Your manuscript is now with our production department. 

With kind regards,

on behalf of

Prof. Vincenzo Lionetti 

Academic Editor

PLOS ONE